# Spectrum Sensing Based on STFT-ImpResNet for Cognitive Radio

**Jianxin Gai \*, Linghui Zhang and Zihao Wei**

The Higher Educational Key Laboratory for Measuring & Control Technology and Instrumentations of Heilongjiang Province, Harbin University of Science and Technology, Harbin 150080, China
\* Correspondence: jxgai@hrbust.edu.cn

**Abstract:** Spectrum sensing is a crucial technology for cognitive radio. The existing spectrum sensing methods generally suffer from certain problems, such as insufficient signal feature representation, low sensing efficiency, high sensibility to noise uncertainty, and drastic degradation in deep networks. In view of these challenges, we propose a spectrum sensing method based on short-time Fourier transform and improved residual network (STFT-ImpResNet) in this work. Specifically, in STFT, the received signal is transformed into a two-dimensional time-frequency matrix which is normalized to a gray image as the input of the network. An improved residual network is designed to classify the signal samples, and a dropout layer is added to the residual block to mitigate over-fitting effectively. We conducted comprehensive evaluations on the proposed spectrum sensing method, which demonstrate that—compared with other current spectrum sensing algorithms—STFT-ImpResNet exhibits higher accuracy and lower computational complexity, as well as strong robustness to noise uncertainty, and it can meet the needs of real-time detection.

**Keywords:** spectrum sensing; residual network; short-time Fourier transform; cognitive radio

## 1. Introduction

With the advent of the 5G era, the lack of spectrum resources has become a realistic problem that is inevitable [1,2]. Spectrum sensing is of vital importance to the optimization of the utilization rate of spectrum resources, and has become the key technology in cognitive radio [3]. In cognitive radio, the secondary user (SU) is allowed to access the spectrum dynamically and randomly without interfering with the primary user (PU) [4]. The main task of spectrum sensing is to explore spectrum holes [5] in order to increase the usage of spectrum resources.

The traditional spectrum sensing methods can be broadly categorized into energy detection (ED) [6,7], matched filter detection [8], cyclostationary feature detection [9], waveform-based sensing [10], and covariance-based detection [11], etc. However, the predefined threshold set by the traditional method has a dramatic influence on the detection probability. With the continuous development of machine learning techniques, the method of realizing spectrum sensing is migrating gradually from conventional statistical methods to machine learning ones. Nowadays, deep learning methods are becoming more and more popular to train spectrum sensing models to classify signals, which improves the detection probability of spectrum sensing, and the model is optimized to approach the pragmatic application level.

At present, some commonly used machine learning methods such as support vector machines (SVM), artificial neural networks (ANN), long-term and short-term memory networks (LSTM), and convolutional neural networks (CNN) have achieved partial success in spectrum sensing. Chen et al. [12] proposed a SVM-based spectrum sensing algorithm to recognize the PU signal by training SVM classifiers based on the energy vectors sampled from SU. Supervised learning and unsupervised learning algorithms such as the naive Bayes classifier, SVM, and hidden Markov model are compared in terms of classification accuracy in [13], in which the experimental results show that the performance of the

SVM algorithm exceeds previous ones. However, as the SVM algorithm uses the time-consuming quadratic programming to solve support vectors, it exhibits high computational complexity in the training process, along with relatively low detection efficiency. Some researchers have proposed new spectrum sensing methods based on ANN and its variants, and also combinations with traditional methods. Tang et al. [14] used energy detection and cyclostationary characteristics to train an ANN model for spectrum sensing, which combines the advantages of energy detection and cyclostationary feature detection while keeping a low computational complexity. The normal likelihood estimation scheme is employed in [15] to input the signal energy detected by the energy detection method to the ANN spectrum sensing model, and the experimental results were better than those provided by the straightforward energy detection methods. In [16], the decision level fusion is introduced to ANN during the spectrum sensing of cooperative users. The decision of each SU achieves the global decision in the fusion center, which improves the detection probability and reduces the false alarm probability in the meantime. However, although the sample data amount increases, the training process is still prone to over-fitting due to the simple design of the network structure, which limits the accuracy of ANN algorithms. To this end, many researchers empower the communication signal recognition tasks by deep learning techniques, regarding signal recognition as a classification problem. Dong et al. [17] extracted both cyclostationary and energy features from noise signals and PU signals, respectively. These features are input to the CNN spectrum sensing model, and the detection can be made by judging whether the frequency band was occupied. However, these features are insufficient to accurately describe the real environment. Subsequently, Pan et al. [18] proposed a cognitive radio spectrum sensing method for orthogonal frequency division multiplexing (OFDM) signals based on the integration of deep learning and the cyclic spectrum. This method analyzed the cyclic auto-correlation characteristics of OFDM signals and the cyclic spectrum obtained by the time domain smoothing fast Fourier transform accumulation algorithm as the input of the CNN model. However, this algorithm is not generalized, as the model is merely tailored to OFDM signals. Wu et al. [19] established a signal modulation recognition model based on CNN-LSTM, which can identify as many as 12 types of signal modulation modes simultaneously. CNN was used to extract the characteristics of the signal space automatically, and then the LSTM network was exploited to extract the time correlation of the extracted signal. The authors in [20] proposed a deep belief network architecture, which achieves better data transmission through the selected path. A spectrum detection network based on deep learning is proposed in [21] to identify the channels; it achieved good results in a low signal-to-noise ratio scenario. Nevertheless, this method is only suitable for specific scenarios, and is not generalized as well. Chen et al. [22] proposed an STFT-CNN spectrum sensing method that utilizes short-time Fourier transform (STFT) to preprocess the signal to make full use of the time-frequency domain information of the signals, and designed a CNN network to classify the signal. This method is a milestone in spectrum sensing. However, the CNN network only contains a single convolution layer, which limits the ability of feature learning. The network's performance can be improved by increasing the network depth, especially for low signal-to-noise ratio (SNR) spectrum signals. However, introducing excessive network layers leads to network degradation. Specifically, the classification accuracy increases with the deepening of the network layers at the beginning. As the network continues to deepen, the accuracy drops sharply after reaching the saturation point. The reason for the network degradation is that with the deepening of the network, the gradient correlation between the shallow network and deep network becomes weak, and a loss of information occurs. To capitulate, traditional spectrum sensing methods have a low utilization rate of signal features with limited feature information extracted, and have a lot of space for the improvement of the accuracy of the spectrum sensing.

In view of the aforementioned issues, we propose a novel spectrum sensing method, named STFT-ImpResNet, based on STFT and an improved residual network (ResNet).

Specifically, the signal samples are preprocessed by STFT, and an improved ResNet (ImpResNet) is designed to classify the signal samples.

The main contributions of our paper are as follows:

- We combined Short-time Fourier transform and a residual network innovatively, and proposed STFT-ImpResNet for spectrum sensing. To the best knowledge of the authors, this is the first time that a combination of STFT and ResNet has been introduced to spectrum sensing.
- We customized a deep learning network structure to achieve a good trade-off between accuracy and computational cost in the context of spectrum sensing. We especially simplified ResNet by replacing the fully connected layer with a global average pooling layer to integrate global information. In addition, a dropout layer was added into the improved residual block to prevent over-fitting effectively.
- We conducted comprehensive experiments which demonstrate that the proposed STFT-ImpResNet algorithm outperforms the existing spectrum sensing algorithms on low signal-to-radio datasets.

The remainder of this paper is organized as follows: In Section 2, we explain the system model of spectrum sensing. The proposed STFT-ImpResNet Spectrum Sensing Algorithm is elaborated in Section 3, followed by the extensive experiments in Section 4 which validate the superiority of STFT-ImpResNet in the balance of accuracy and detection efficiency. Finally, we conclude this work in Section 5.

## 2. System Model

In cognitive radio networks, spectrum sensing can be transformed into a binary hypothesis test problem. The binary sensing model can be represented as follows:

$$
\begin{aligned}
H_0 &: x(n) = w(n) \\
H_1 &: x(n) = s(n) + w(n)
\end{aligned}
\tag{1}
$$

where $x(n)$ represents the received signal, $s(n)$ denotes the signal emitted by PU, $\omega(n)$ is the noise, and $H_0$ and $H_1$ represent the hypotheses of signal-containing noise only, and that containing both the signal emitted by PU and noise, respectively.

Detection probability $P_d$ and false alarm probability $P_f$ are essential indicators for the evaluation of the spectrum sensing model, and are described by the following equation:

$$
\begin{aligned}
P_d &= P\{H_1|H_1\} \\
P_f &= P\{H_1|H_0\}
\end{aligned}
\tag{2}
$$

## 3. STFT-ImpResNet Spectrum Sensing Algorithm

Essentially, STFT implements a universal time-frequency domain transformation that is infeasible for any deep learning networks to learn based on limited datasets. From this viewpoint, STFT is equivalent to a deep learning network layer transformation based on infinite signal samples from the real world. By adding STFT as the first step before the ResNet network, the whole STFT-ImpResNet method becomes equivalent to combining a universal rule-based network layer and data-dependant neural network layers into a composite one. Therefore, the resulting STFT-ImpResNet will have better learning ability due its greater number of network layers, whilst also being less prone to over-fitting because of the universal characteristics of STFT. In other words, STFT-ImpResNet is equivalent to fine-tuning the universal time-frequency transformation network using ResNet. Because fine-tuning universal network models often produces superior performance for general classification tasks [23,24], while niche approaches succeed using networks trained from scratch, we designed this STFT-ImpResNet structure.

### 3.1. Short-Time Fourier Transform

Most of the existing spectrum sensing methods based on deep learning generate two-dimensional gray images by directly truncating and splicing one-dimensional received signals. This method is simple yet efficient, but lacks signal frequency domain information. In order to further improve the performance of the spectrum sensing model under low SNR, and to extract the time-frequency information of the received signal more effectively, this paper utilizes short-time Fourier transform to preprocess the signal. Two-dimensional gray images are generated by the time-frequency analysis of the received one-dimensional signals, which can fully reflect the frequency domain characteristics of the signal, and are supposed to exhibit strong robustness and noise immunity [25].

STFT decomposes the one-dimensional time-domain signal into equal-length short segments implemented by a window function. Then, discrete Fourier transform (DFT) is performed on each short segment in order to obtain the spectrum signals.

The short-time Fourier transform of discrete signals is defined as

$$STFT_x(t,\omega) = \sum_{n=-\infty}^{L-1} x(n)g(n-t)e^{-jm\omega} \tag{3}$$

$g(n)$ represents the window function, and the selection of different window functions will also bring some differences. A rectangular window has the highest frequency resolution, but the spectrum leakage is serious. Compared with a rectangular window, a Hanning window has a stronger ability to reduce spectral leakage, but it is poor in frequency resolution compared with a Hamming window [26]. A Hamming window is adopted in this proposed method, as since it has good performance in frequency resolution and reducing spectrum leakage. However, the analysis of signal characteristics will not produce significant differences with the change of window function.

The spectrogram of the signal is expressed as follows:

$$SP_x(t,\omega) = |STFT_x(t,\omega)|^2 \tag{4}$$

Using the spectrum sensing model, the received signal sample is simplified as follows:

$$X = [x(1), x(2), \cdots x(n)] \tag{5}$$

The plural matrix is obtained by the STFT transformation of the received signal sample *X*. The matrix is normalized to generate a gray image as follows:

$$X_s = \left[x_S{}^{(1)}, x_S{}^{(2)}, \cdots x_S{}^{(n)}\right] \tag{6}$$

### 3.2. Improved ResNet

As early as 2016, He et al. [27] proposed ResNet, constructing residual blocks by adding identity mappings as a shortcut connection to CNN. ResNet overcomes the problems of network convergence and network degradation caused by increasing the number of network layers in CNN [28]. He et al. compared the training error and test error of 20-layer and 56-layer "plain" networks on CIFAR-10. When the network is deepened from 20 layers to 56 layers, the test error is expected to decrease in theory. However, it is evident from the experimental results that the test error increases while the training error also increases as the depth of the network layer grows. Therefore, the reason for the increase of the test error is not over fitting, but network degradation. To solve this issue, the modification of the network structure is more viable compared to simply increasing the network depth.

The structure of the standard residual block is shown in Figure 1a. Given network input $x$ and the expected output $H(x)$, if $x$ is directly transmitted to an intermediate result, then the output through the convolution layer is $F(x) = H(x) - x$, rather than learning the output $H(x)$ without shortcut connections. The optimization of the former

is simpler than that of the latter because the residual feature output map is the first order difference of output and input signals, resulting in a one-order smaller magnitude than the straightforward output. Specially, when the output of the convolution layer is zero, the identity mapping $H(x) = x$ can be obtained.

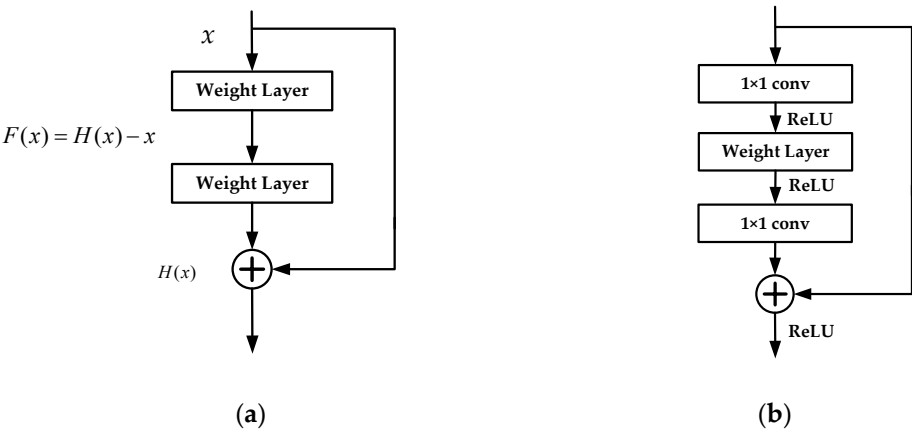

**Figure 1.** Residual block: (**a**) standard residual block; (**b**) residual bottleneck block.

The residual block designed in this work is the residual bottleneck block. As compared with the standard residual block, the bottleneck residual block adds $1 \times 1$ convolution before and after the feature extraction convolution layer, which requires less input dimension of the weight layer, and thus reduces the amount of parameter calculation. The structure of the residual bottleneck block is depicted in Figure 1b.

The structure of the improved residual block (ImpRB) designed in this work is illustrated in Figure 2. In ImpRB, the ReLU activation function is interposed after each convolution layer, and the three linear regression activation functions have a stronger ability to extract feature information than a single linear regression activation function. In order to further improve the extraction efficiency of the network, batch normalization (BN) is introduced, which accelerates the convergence process of the network, rendering a more robust training of the whole network. With limited training data, the spectrum sensing model is prone to over-fitting, especially when deepening the network. Therefore, we additionally integrate the dropout layer in the proposed ImpRB structure.

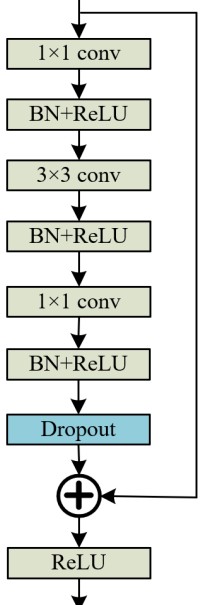

**Figure 2.** Improved residual block.

The principle of dropout is briefed as follows: some hidden nodes are randomly ignored during the training process in order to reduce the number of intermediate features, and a new hidden layer is constructed. Therefore, the network model of each training batch is different, which increases the orthogonality of each layer to prevent over-fitting, and improves the generalization ability of the model. Moreover, the dropout layer only participates in the network training process instead of the network testing process, such that it will not affect the detection efficiency. The network models of the standard network and dropout network are shown respectively in Figure 3a,b.

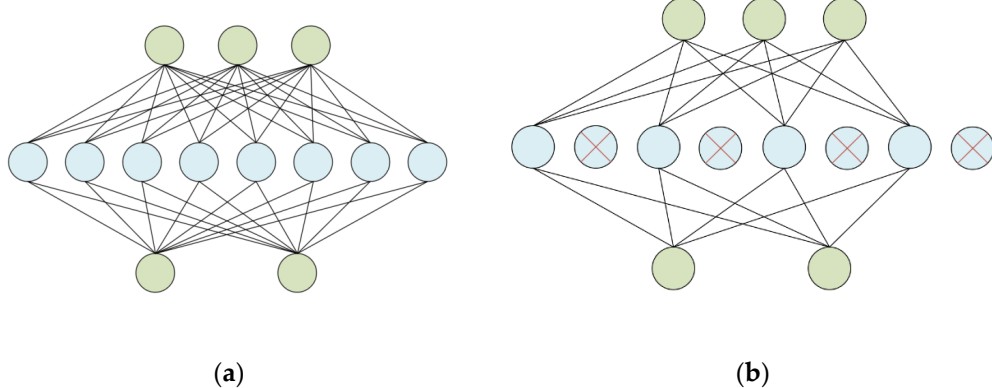

(**a**)                      (**b**)

**Figure 3.** Comparison of the standard network and dropout network: (**a**) standard network; (**b**) dropout network with dropout rate of 0.5.

The feed-forward operation of a standard neural network is represented as follows:

$$z_i^{(l+1)} = w_i^{(l+1)} y_i^l + b_i^{(l+1)}$$
$$y_i^{(l+1)} = f(z_i^{(l+1)}) \tag{7}$$

When the dropout finishes, the expression of the dropout network becomes

$$r_j^{(l)} \sim Bernoulli(p)$$
$$\widetilde{y}^{(l)} = r^{(l)} * y^{(l)}$$
$$z_i^{(l+1)} = w_i^{(l+1)} \widetilde{y}^{(l)} + b_i^{(l+1)}$$
$$y_i^{(l+1)} = f(z_i^{(l+1)}) \tag{8}$$

where $r_j^{(l)}$ is the 0/1 vector randomly generated by the Bernoulli function and used as an activation function of neuron $y^{(l)}$, in which some neurons are randomly set to 0 by setting dropout rates; $\widetilde{y}^{(l)}$ represents the neuron after dropout; $z_i^{(l+1)}$ indicates the neuron to be activated in layer $l + 1$; $y_i^{(l+1)}$ is the output neuron in layer $l + 1$; $f(\cdot)$ is the activation function; and $w_i^{(l+1)}$ and $b_i^{(l+1)}$ denote the weight and bias of layer $l + 1$ respectively.

When the number of input and output channels is equal, the shortcut connection is identity mapping, and the output of the residual module is

$$H(x) = f(z) + x \tag{9}$$

The output of layer $l$ of the residual network is

$$H_l(x) = f(z_l) + x_l \tag{10}$$

In order to train the network, the signal samples in $X_s$ are labeled as

$$Y = \{(x_S^{(1)}, y^{(1)}), (x_S^{(2)}, y^{(2)}), \cdots (x_S^{(n)}, y^{(n)})\} \tag{11}$$

In the above formula, $n$ is the number of samples in the dataset and $y^{(\cdot)}$ is the spectrum status label, $y^{(\cdot)} \in (0,1)$. When the spectrum is idle, $y^{(\cdot)} = 0$; when the spectrum is occupied, $y^{(\cdot)} = 1$. The ImpResNet structure is shown in Figure 4.

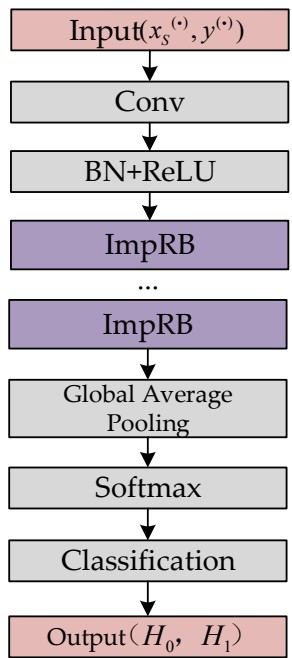

**Figure 4.** ImpResNet structure in the spectrum sensing model.

The signal samples pass through one-dimensional convolution first, and then enter the ImpRB. Multiple improved residual blocks are superimposed to enhance the ability of feature extraction. After residual learning, the fully connected layer (FC) is replaced by global average pooling (GAP) [29]. GAP was chosen because it takes the average of each feature map which can integrate global spatial information. Compared with FC, GAP does not require a large number of training parameters, thereby reducing the risk of over-fitting. The output dimensions of the GAP are equal to the number of categories, and thus the resulting vector can be fed directly into the Softmax layer, which maps the outputs of multiple neurons into the interval of (0,1) to realize the classification of samples.

The input-output mapping of the whole network is

$$F_D(x_i, \{W\}) = \hat{y}^{(i)} \cong y^{(i)} \tag{12}$$

Cross-entropy is adopted as a loss function in the algorithm, formulated as follows:

$$L = -\left[ \sum_{i=1}^{N} y^{(i)} \log \hat{y}^{(i)} + (1 - y^{(i)} \log(1 - \hat{y}^{(i)}) \right] \tag{13}$$

The training dataset selects $m$ pairs from the training dataset:

$$Y_{training} = \{(x_S^{(1)}, y^{(1)}), (x_S^{(2)}, y^{(2)}), \cdots (x_S^{(m)}, y^{(m)})\} \tag{14}$$

The test dataset adopts $n$-$m$ pairs of test data:

$$Y_{test} = \{(x_S^{(m+1)}, y^{(m+1)}), (x_S^{(m+2)}, y^{(m+2)}), \cdots (x_S^{(n)}, y^{(n)})\} \tag{15}$$

The spectrum sensing algorithm based on STFT-ImpResNet is elaborated in Algorithm 1.

---

**Algorithm 1:** STFT-ImpResNet Spectrum Sensing

---

**Input:** Received signal samples $X$; maximum number of iterations *IterMax*
**Output:** $P_d$ and $P_f$
1.  Preprocess the received signal samples $X$ into training dataset $Y_{training}$ and test dataset $Y_{test}$
2.  Initialization: iteration counter $i = 0$ and random weight $W$
3.  Gradient descent training of training dataset in ImpResNet
    **Repeat**
        Update $\hat{y}_{(i)}$ according to Equation (12);
        Substitute $\hat{y}_{(i)}$ into Equation (13).
    **Until** *IterMax* reached
4.  Inference: apply the trained ImpResNet model to the online dataset and output the classification results
5.  Calculate the detection probability and the false alarm probability

---

## 4. Experimental Result Analysis

### 4.1. System Simulation

The training dataset and test dataset in this work are generated by QPSK system simulation. The transmitted signal is modulated by the in-phase carrier and orthogonal carrier after serial-parallel conversion, and the QPSK signal is obtained after addition. The QPSK signal is transmitted to users and demodulated after going through the Rayleigh channel and white Gaussian noise. During demodulation, the QPSK signal forms two identical signal channels through the power separator, and then a coherent demodulation is performed. After the sampling decision and parallel-series conversion, $I$ and $Q$ signals are obtained. The signal samples are transformed by STFT to generate datasets for network training. The simulation flow of the QPSK system is summarized in Figure 5.

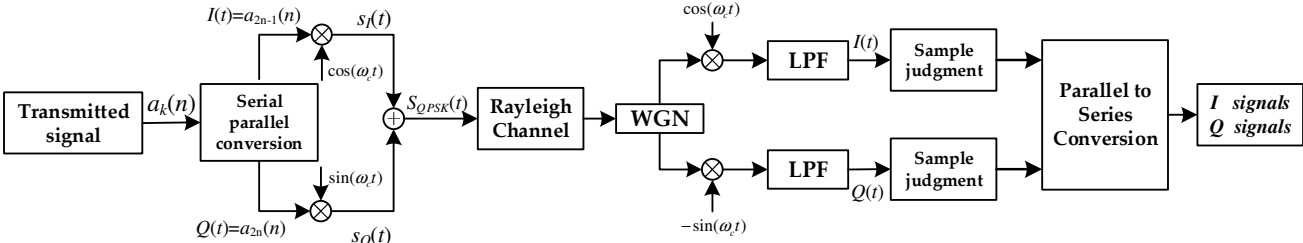

**Figure 5.** Simulation flow of the QPSK system.

The datasets are generated with the sampling length of 512, with a window length of 32 and a carrier frequency of 40 kHz. The SNR range of the mixed-SNR dataset is between $-25$ and $-1$ dB at the interval of 2 dB. Under each SNR sample, 100 samples are generated as the training dataset and 50 samples are generated as the validation dataset. The low-SNR dataset is fixed at $-19$ dB, where 1000 received signal samples are collected for training purposes and 100 received signal samples are collected for validation purposes. The noise is white Gaussian noise (WGN) with the noise power of 1 dBW, and the number of noise samples generated at each SNR is kept the same as the number of signals.

### 4.2. Experimental Configuration

In order to verify the performance of the STFT-ImpResNet spectrum sensing algorithm proposed in this work, simulation experiments were carried out. The simulation environments are given as follows: MATLAB R2020b, Intel® Core® CPU i5-11300H and NVIDIA® GeForce MX450. We used stochastic gradient descent (SGD) with a mini-batch size of 128, and the initial learning rate was set to 0.001.

### 4.3. Experimental Results

In the following experiments, we adopt network accuracy commonly used by scholars to evaluate the performance of algorithms, where the accuracy refers to the proportion of "Ture Positive" and "Ture Negative" in all classification cases in spectrum sensing [30].

#### 4.3.1. Effects of Dropout Rates

Given dropout rates of 0, 0.1, 0.2, 0.3, 0.4, 0.5, 0.6, and 0.7, which are feasible settings in practice, the network accuracy of different dropout rates was tested under the mixed-signal SNR dataset.

The experimental results shown in Table 1 show that when the network is the basic residual network and the dropout rate is 0, the accuracy in the training dataset is 100%. However, due to the large number of network parameters, over-fitting occurs and the validation accuracy is only 93.2%. Compared with the baseline network, the validation accuracy of the network with dropout is improved. However, when the dropout rate is large, there are too many lost neurons, resulting in a decrease of validation accuracy, implying an under-fitting phenomenon. When the dropout rate is 0.4, the validation accuracy reaches the maximum of 99.4%, which not only prevents the network from over-fitting but also prevents the sample from losing the most representative features.

**Table 1.** Classification accuracy of different dropout rates.

| Dropout Rate | Training Accuracy (%) | Validation Accuracy (%) |
|:---:|:---:|:---:|
| 0 | 100 | 93.2 |
| 0.1 | 100 | 95.7 |
| 0.2 | 100 | 97.7 |
| 0.3 | 100 | 98.5 |
| 0.4 | 100 | 99.4 |
| 0.5 | 100 | 98.2 |
| 0.6 | 100 | 96.4 |
| 0.7 | 100 | 94.3 |

#### 4.3.2. Effect of the Network Structure

In this experiment, we compare the spectrum sensing model accuracy of ResNet with different structures and CNN. The number of kernels in the residual network and the size of the convolution kernels in the residual module are changed to construct different residual network structures. On the mixed-SNR dataset, we select the residual network with the number of filters of 8 and 10, and the sizes of convolution kernel of $3 \times 3$ and $5 \times 5$, and the classification accuracy of networks with five different structures, as shown in Figure 6. The network parameters of ResNet1, ResNet2, ResNet3, ResNet4 and CNN are listed in Table 2.

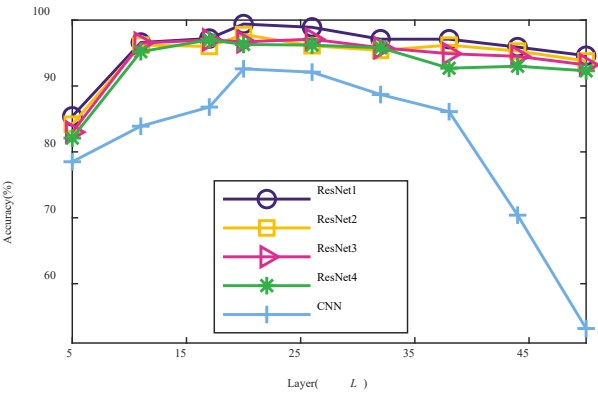

**Figure 6.** Effect of the network layers on accuracy.

**Table 2.** Network parameter.

|  | **Number of Kernels** | **Kernel Size** | **20-Layer Accuracy** |
|---|---|---|---|
| ResNet1 | 8 | $3 \times 3$ | 99.4% |
| ResNet2 | 8 | $5 \times 5$ | 97.8% |
| ResNet3 | 10 | $3 \times 3$ | 96.7% |
| ResNet4 | 10 | $5 \times 5$ | 96.3% |
| CNN | 8 | $3 \times 3$ | 92.6% |

It is evident that the accuracy of the four residual networks is higher than that of the baseline CNN with the same number of network layers. For instance, when the network depth is 20, the classification accuracy of ResNet with four structures is 6.8%, 5.2%, 4.1% and 3.7% higher than that of CNN, respectively. The accuracy of CNN tends to saturate as the network depth grows, and there is a drastic degradation after the network exceeds 26 layers. The accuracy of CNN is as low as 53.2% at 50 layers, while the accuracies of ResNets with four structures are 41.4%, 40.6%, 40.0% and 39.1% higher than CNN, respectively. Because CNN adopts a large number of activation functions in the process of network deepening, the original data is transformed nonlinearly, which makes the model unable to realize linear transformation. Therefore, the introduction of identity mapping is particularly necessary.

By comparing the accuracy of ResNet with different structures, it can be seen that the increase of the filter number and convolution kernel size will lead to lower accuracy eventually. This is because increasing the number and size of convolution kernels will increase the calculation of parameters. It can be seen that Resnet1 has the highest classification accuracy of 99.4% on the layers of 20. Therefore, the spectrum sensing model proposed in this work selects the ResNet1 network with the 20 network layers.

### 4.3.3. Effect of Sampling Points

This subsection compares the change of detection probability along with SNR of the STFT-ImpReNet spectrum sensing model with various numbers of sampling point $N$. The experimental results are shown in Figure 7.

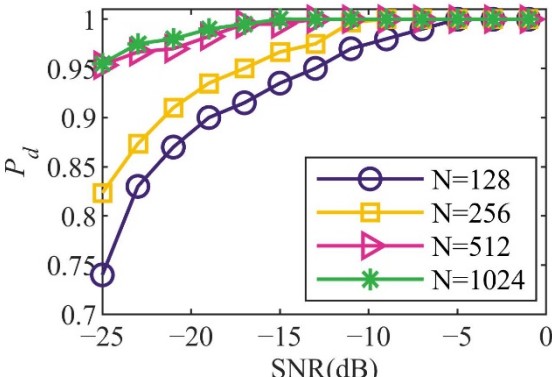

**Figure 7.** Effect of sampling points.

The experimental results show that the more sampling points there are, the higher the detection probability of the algorithm. However, the number of sampling points will also affect the detection efficiency. The larger the number of sampling points is, the longer the training time and detection time are. The growth trend is not obvious when the number of sampling points reaches 512 and continues to increase. This is because excessive sampling points will reduce the relevance of the information before and after sampling, and it is difficult for the network to learn more effective correlation information. Based on preliminary experiments, the sampling point $N$ of the spectrum sensing model proposed in this work is set to be 512 as the optimum.

### 4.3.4. Comparison of Efficiency

The proposed STFT-ImpResNet, STFT-CNN [22] and SVM [12] methods all need network training in order to establish a spectrum sensing model before detection, while the ED [7] method can detect signals directly without training. The detection efficiency of four methods is compared under the low SNR dataset (see Section 4.1 for the generation of the dataset), and the results in Table 3 report the detection probability $P_d$, the false-alarm probability $P_f$, training time and detection time of different methods, respectively.

**Table 3.** Comparison of the detection efficiency.

|  | $P_d$ | $P_f$ | Training Time (s) | Detection Time (ms) |
|---|---|---|---|---|
| STFT-ImpResNet | 0.99 | 0.04 | 29.186 | 12.52 |
| STFT-CNN | 0.91 | 0.3 | 33.143 | 13.95 |
| SVM | 0.6 | 0.42 | 15.646 | 19.03 |
| ED | 0.36 | 0.33 | - | 2.93 |

The offline training time and online detection time of the STFT-ImpResNet spectrum sensing model are shorter than those of CNN, because the identity mapping in the ImpResNet accelerates the convergence speed of the network and shortens the training time. Due to the low computational complexity of the ImpResNet algorithm, the detection time of STFT-ImpResNet is shorter than that of SVM.

The computational complexity of SVM is $O(n + n^3)$ [31], and the computational complexity of CNN is $O(n\sum_{l=1}^{L} F_l^2 K_l^2 Q_l Q_{l-1})$, where $L$ represents the number of network layers, $F_l$, $K_l$, and $Q_l$ are the size of the output feature map, the size of the convolution kernel, and the number of output channels in the layer $L$, respectively.

ImpResNet adds identity mapping to CNN, but the identity mapping neither adds additional parameters nor increases computational complexity. In the meanwhile, the identity mapping can directly connect different layers, while the computational complexity of CNN needs to be accumulated in each layer. Therefore, the order of algorithm complexity of the four methods is SVM > CNN > ImpResNet > ED.

Compared with SVM and ED, STFT-ImpResNet needs a longer training time, but the training process takes only one time and does not need to be updated frequently. In addition, our algorithm takes only 12.52 ms to carry out an inference of a sample, which can meet the needs of real-time detection.

### 4.3.5. Effect of Noise Uncertainty

In this subsection, we compare the detection probability of datasets processed by traditional methods (cut and splice) [32] with those after STFT under different noise power. Introducing $a$ into noise model as noise uncertainty coefficient, such as $a \geq 1$, if the noise power does not fluctuate, the noise uncertainty coefficient is 1 according to [33]. In order to further illustrate the robustness of the proposed algorithm to noise uncertainty, extra experiments were conducted, and the noise considered in the experiment is additive white Gaussian noise (AWGN).

The results in Figure 8 show that when SNR = $-19$ dB and the noise power is 1 dBW, 1.2 dBW and 1.5 dBW, the detection probabilities of STFT-ImpResNet are 0.945, 0.945 and 0.94, respectively, which implies high robustness to the noise uncertainty. Meanwhile, the detection probabilities of the truncating and splicing spectrum sensing model are 0.71, 0.675, and 0.635, respectively, which are greatly affected by the noise power. STFT can effectively suppress noise and improve SNR. Therefore, time-frequency analysis of signal samples with STFT is not only able to improve the detection probability of the system but also improve the robustness of the spectrum sensing model to noise uncertainty.

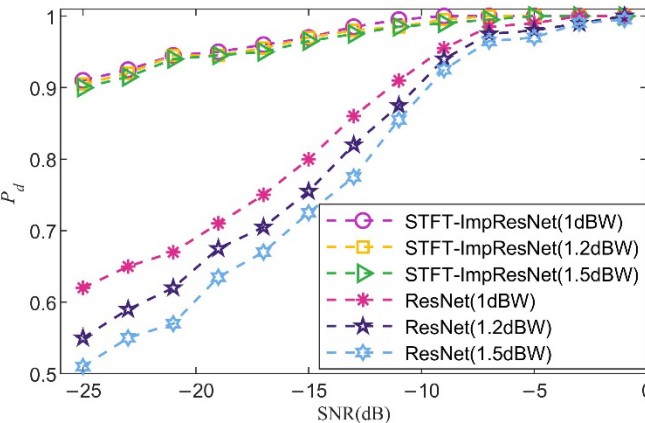

**Figure 8.** Detection probability curve of different noise powers.

4.3.6. Comparison of ROC

In order to further evaluate the detection performance, we aligned corresponding $P_d$ and $P_f$ pairs in all of the 1000 spectrum sensing cases, forming the Receiver Operating Characteristics (ROC) curves by comparing the proposed STFT-ImpResNet method with the STFT-CNN [22], SVM [12] and ED [7] under $-19$ dB. As shown in Figure 9, the proposed STFT-ImpResNet algorithm exhibits a higher detection probability than the other three algorithms under the same $P_f$. The detection probability of STFT-ImpResNet is 0.945 when $P_f = 0.01$, which is 0.17, 0.565 and 0.884 higher than STFT-CNN, SVM and ED, respectively. Given $P_d = 1$, the false-alarm probability of STFT-ImpResNet and STFT-CNN is 0.2 and 0.8, respectively. The detection probability of STFT-ImpResNet is the highest among the four algorithms given any $P_f$. It can be safely concluded that the proposed STFT-ImpResNet outperforms the performance of state-of-the-art methods on real scenario datasets.

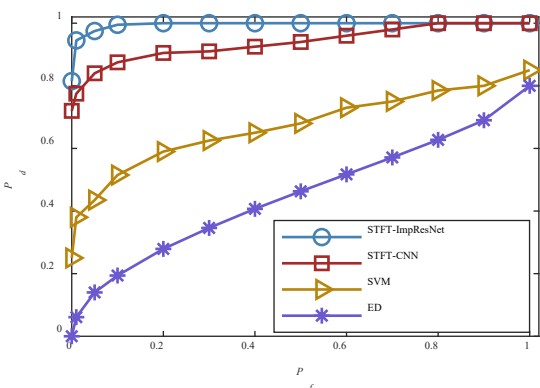

**Figure 9.** ROC curves of different algorithms.

**5. Conclusions**

In this work, we innovatively incorporate STFT and ResNet into a spectrum sensing neural network. The traditional feature extraction method relies on a one-dimensional signal, ignoring the time-frequency characteristics, which limits feature extraction ability. The proposed method has higher robustness to noise uncertainty than the traditional one. In order to prevent over-fitting, the dropout layer is also added to the design of the residual block, together with global average pooling instead of a fully connected layer to integrate global information. In the meantime, the ImpResNet designed in this paper can effectively prevent the degradation of CNN under a deep network. The experimental results show that the proposed STFT-ImpResNet remarkably outperforms state-of-the-art spectrum sensing methods in terms of detection probability, false-alarm probability and detection efficiency, and the proposed algorithm achieves an excellent trade-off between accuracy and

efficiency. Future work may include exploring the role of residual networks in cooperative spectrum sensing.

**Author Contributions:** Conceptualization, J.G.; methodology, J.G.; software, L.Z.; validation, J.G., L.Z. and Z.W.; formal analysis, L.Z.; investigation, Z.W.; resources, J.G.; data curation, L.Z.; writing—original draft preparation, L.Z.; writing—review and editing, J.G.; visualization, Z.W.; supervision, J.G.; project administration, J.G.; funding acquisition, J.G. All authors have read and agreed to the published version of the manuscript.

**Funding:** This research was funded by National Natural Science Foundation of China (No. 61501150), Natural Science Foundation of Heilongjiang Province (No. QC2014C074), and Fundamental Research Funds for the Universities in Heilongjiang Province (No. 2018-KYYWF-1656).

**Institutional Review Board Statement:** Not applicable.

**Informed Consent Statement:** Not applicable.

**Data Availability Statement:** The data used to support the findings of this study are included within the article.

**Conflicts of Interest:** The authors declare no conflict of interest.

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
