# Peer review of "Spectrum Sensing Based on STFT-ImpResNet for Cognitive Radio"

_electronics, doi:10.3390/electronics11152437_

Round 1

Reviewer 1 Report

The following are my review comments-

1. Table 1 is not available in the entire manuscript.

2. All mathematical equations must be cited. 

3. Authors must show the comparative study with the latest techniques.

4. Abstract and Conclusion section must be impressive and effective.

5. I still see some missing literature review, authors must cite some more papers, and papers are given as

(i) Simulation of a Smart Sensor Detection Scheme for Wireless Communication Based on Modeling, Electronics journal, MDPI, vol. 9, No., page-1506, 2020.

(ii) Performance Analysis of Deep Learning-Based Routing Protocol for an Efficient Data Transmission in 5G WSN Communication, IEEE ACCESS 2022 (2022), 9340-9356.

(iii) Improved sensing detector for wireless regional area networks, The International Journal of Cogent Engineering, Taylor and Francis, Vol. 4, No. 1, January 2017.

I appreciate authors work, there are some suggestions to be incorporated, kindly submit again.

Reviewer 2 Report

The authors propose a spectrum sensing approach based on a combination of short-time Fourier transform and an improved residual network. The performance of the proposed method is verified by means of simulations and is compared with the one of other existing algorithms. Several performance aspects (sensing time, computational complexity, probability of detection) are taken into account. The paper is clearly written and the results that are obtained are promising. The authors should take into account the following comments and suggestions in order to improve the overall quality of the paper.

-          It is not clear what the drastic degradation in deep network that is mentioned in the abstract is referring to, in the context of spectrum sensing methods; (rows 11-12);

-          …lower computational… instead of …low computational… (row 19);

-          There are also other categories of spectrum sensing methods other then the three that are mentioned on row 31;

-          The ResNet abbreviation should be explained the first time when in it used (row 95); other abbreviations like SNR (row 137) are also not explained;

-          A remainder presenting the contents for the rest of the paper would be useful at the end of section 1;

-          Represent instead of represents (row 115) and containing instead of contains (row 116);

-          It should be explicitly explained what is meant by some differences (row 148);

-          Denote instead of denotes (row 207);

-          The accuracy that is used as metric in sections 4.3.1-4.3.2 should be defined;

-          It would be useful to include a figure in which the variation of the accuracy with the number of layers is represented (regarding the comments from rows 288-293);

-          In figures 8 and 9 the abbreviation STST-ImpResNet should be used to refer to the proposed method;

-          A reference regarding the traditional methods (cut and splice) mentioned ar rows 350-351 should be provided;

-          What is the actual SNR range of the low SNR dataset mentioned at row 326?

Round 2

Reviewer 1 Report

Author(s) have incorporated all the suggestions in the revised manuscript. Paper is accepted in its present form.